# Cardiovascular Magnetic Resonance Imaging Pattern in *Campylobacter jejuni*-related Myocarditis

**DOI:** 10.3390/microorganisms10020208

**Published:** 2022-01-19

**Authors:** Nabil Belfeki, Souheil Zayet, Mohannad Yassin, Mazen Alloujami, Audrey Lefoulon, Théo Pezel, Jerôme Garot, Cyrus Moini

**Affiliations:** 1Department of Internal Medicine, Groupe Hospitalier Sud Ile de France, 77000 Melun, France; belfeki.nabil@gmail.com; 2Department of Infectious Diseases, Hôpital Nord Franche-Comté, 90400 Trévenans, France; 3Department of Cardiology, Groupe Hospitalier Sud Ile de France, 77000 Melun, France; mohannad.yassin@ghsif.fr (M.Y.); mazen.alloujami@ghsif.fr (M.A.); audrey.lefoulon@ghsif.fr (A.L.); cyrus.moini@ghsif.fr (C.M.); 4Department of Cardiovascular Imaging, Hôpital Jacques Cartier, Ramsay Santé, 91300 Massy, France; theo.pezel@gmail.com (T.P.); jerome.garot@gmail.com (J.G.)

**Keywords:** *Campylobacter jejuni*, enteritis, myocarditis, CMR imaging

## Abstract

Background: *Campylobacter jejuni* (*C. jejuni*) is a common cause of mostly self-limiting enterocolitis. Although rare, myocarditis has been increasingly documented as a complication following campylobacteriosis. Such cases have occurred predominantly in younger males and involved a single causative species, namely *C. jejuni*. Case report: We report herein a case of myocarditis complicating gastroenteritis in a 23-year-old immunocompetent patient, caused by this bacterium with a favorable outcome. Cardiac magnetic resonance imagining was useful in establishing an early diagnosis. Conclusions: Myocarditis should be considered in younger patients presenting with chest pain and plasmatic troponin elevations. The occurrence of myocarditis complicating *C. jejuni* is reviewed.

## 1. Background

*Campylobacter* spp. are among the most common bacterial causes of gastroenteritis [1]. *Campylobacter* genus encompasses many species, among which *C. jejuni, C. coli* and *C. fetus* are the main human pathogens [1]. Typical symptoms include GI symptoms and fever in elderly and immunocompromised individuals and those with occupational exposure to infected animals. *C. jejuni* are a well-recognized but rare cause of myocarditis and perimyocarditis [2,3,4,5,6]. We report herein a case of myocarditis with no bacteremia, due to *C. jejuni*, occurring in an immunocompetent patient. The diagnosis was based on clinical, biological and imaging findings.

## 2. Case Presentation

A 23-year-old male patient without past medical history presented to the emergency room with recent retrosternal chest tightness and pain. He reported recent gastro-intestinal (GI) symptoms of nausea, vomiting, 4-day watery diarrhea with transient episode of fever (38.7 °C). At admission, physical examination showed an impaired general status. He was afebrile (37.2 °C), respiratory rate 20 per min., cardiac assessment showed blood pressure of 140/70 mmHg, regular tachycardia of 100 beats per minute and normal cardiac murmur. Pulmonary examination showed a respiratory rate of 16 cycles per min, and auscultation was normal. He denied abdominal pain, and palpation showed no tenderness and no liver or spleen enlargement. He reported generalized myalgia but rheumatological evaluation did not show arthritis or productive myalgia. The electrocardiogram (ECG) showed left axis deviation with regular sinusal tachycardia. Routine laboratory showed leukocytosis 13 G/L (normal range < 10 G/L), neutrophilia 8.2 G/L (normal range 1.5–7 G/L) and lymphopenia 600/mm^3^ (normal range 1500 to 4000/mm^3^) on cell blood count, with elevated C-reactive protein of 130 mg/dL (normal range < 5 mg/dL), serum electrolytes, creatinine and liver enzymes within normal limits. Maximum Troponins and creatine kinase MB (CK-MB) were elevated to 678 ng/L (normal range < 14 ng/L) and 54 ng/mL (normal range < 7 ng/mL). The patient was admitted to the cardiac department for further workup. He was managed symptomatically with analgesics, anti-reflux and fluids. Transthoracic echocardiography (TTE) revealed a preserved ejection fraction (EF) of 55% with normal wall motions, no valvular dysfunction, normal pulmonary pressure and no pericardial effusion. His risk of coronary artery disease (CAD) was low; moreover, clinical, biological and echocardiographic presentation summed the hypothesis of an acute myocarditis. A large etiological workup, including repeated peripheral blood culture, *Mycoplasma pneumoniae, Chlamydia pneumoniae, Coxiella burnetii*, *Borrelia burgdorferi, Leptospira* spp., *Rickettsia* spp. and *Brucella* spp., was conducted. The serologies, such as urinary *Legionella pneumophila* antigen, were negative. Moreover, serologies of RNA viruses (coxsackieviruses A and B, hepatitis C virus, human immunodeficiency virus) and DNA viruses (adenoviruses, parvovirus B19, cytomegalovirus, human herpes virus-6, Epstein-Barr virus, varicella-zoster virus and herpes simplex virus) were negative. Autoimmune assessment, including antinuclear, anti-neutrophil cytoplasmic antibodies, systemic sclerosis and autoimmune myopathies specific antibodies and converting enzyme assay, were negative. The patient denied any recent drug intake. A Gram stain of specimen stool collected showed multiple curved and spiral Gram-negative rods. Biochemical tests indicated an oxidase, catalase and hippurate negative and indoxyl acetate-positive bacterial species, corresponding to *C. jejuni.* Stool cultures confirmed the diagnosis of *C. jejuni* sensitive to macrolides (Azithromycin/Roxithromycin/Clarythromycin) and flouroquinolones (Ciprofloxacin). Continuous telemetry monitoring showed some runs of non-sustained ventricular tachycardia (NSVT). Oral bisoprolol 2.5 mg twice daily was started for that, and oral 1 g of Azithromycin was administered. He remained clinically stable over the rest of the hospital course, and the diarrhea was progressively resolved. The patient remained stable, and we could perform cardiovascular magnetic resonance (CMR) imaging. Triple inversion-recovery black-blood T2-weighted STIR sequences showed focal areas of hypersignal in the subepicardium of the posterolateral left ventricular (LV) wall, indicative of myocardial edema (Figure 1). In addition, steady-state-free-precession (SSFP) cine CMR showed early hypersignal in the subepicardium of the posterolateral LV wall immediately after injection of 0.1 mM of Gadolinium chelates, indicating focal hyperemia (Figure 2). Inversion-recovery gradient-echo-based late Gadolinium enhancement techniques, acquired 10 min. after Gadolinium injection, revealed subepicardial nodular lesions of myocardial damage (Figure 3). The final diagnosis of *C. jejuni*-related acute myocarditis was supported by the Lake Louise criteria [7]. The patient was discharged free of symptoms after one week in hospital. On close follow-up, his C-reactive protein and cardiac enzymes normalized after three weeks. Repeated TTE and 24-h ECG were normal, so bisoprolol was progressively discontinued after 6 months. Control CMR imaging at 3 months showed regression of the focal areas of hyper signal in the sub epicardium of the posterolateral left ventricular (LV) wall.

## 3. Discussion and Conclusions

This is a rare case of an acute myocarditis related to *C. jejuni* infection in a young male patient. Myocarditis may have different patterns of clinical presentation, ranging from mild symptoms of chest pain and tightness associated with transient ECG changes, as illustrated in the current case, to life-threatening cardiogenic shock and ventricular arrhythmia. Thus, it is of high importance to evoke this diagnosis and carry out appropriate investigation to identify its cause [8]. The diagnosis of myocarditis is established by a combination of clinical, laboratory and cardiac imaging criteria. The diagnosis should be considered in patients with unexplained cardiac symptoms, with elevated cardiac enzyme levels in the absence of coronary artery disease. ECG can show arrhythmias, atrioventricular block, non-specific repolarization abnormalities, but it could be normal, as illustrated in the present case [2]. TTE must be performed at presentation in the management of patients with suspicious of myocarditis and may rule out non-inflammatory cardiac disease. Unspecific imaging findings include global ventricular dysfunction, regional wall motion abnormalities and diastolic dysfunction with preserved ejection fraction and may occur in myocarditis. Moreover, the place of Doppler tissue or strain-rate imaging in the diagnosis of myocarditis remains to be determined [9]. TTE can be normal at presentation and should be repeated during hospitalization if there is any worsening of hemodynamics. This means that diagnosing acute myocarditis is a really challenging task and probably suffers from a lack of a gold standard test. However, CMR imaging provides non-invasive tissue characterization of the myocardium and can support the diagnosis of myocarditis, according to Lake Louise Criteria [7]. In the present case, CMR imaging performed during the inpatient stay showed normal left volumes and function, with localized myocardial oedema and contrast enhancement within the basal inferolateral wall consistent with acute myocarditis. To the best of our knowledge, a total number of 45 clinical case in the English literature of *C. jejuni*-related myocarditis have been reported [3]. CMR imaging was not reported in all previous cases because it would depend upon local availability and expertise. In our case, CMR imaging was helpful in establishing the diagnosis and avoided the need for coronary angiography or invasive cardiac biopsy, as the risks outweighed any potential diagnostic benefit. In addition, we observed a relatively benign course to this illness, with only supportive care required. Moreover, CMR imaging showed subepicardial nodular lesions after gadolinium injection. This finding has not been reported specifically in campylobacter-related myocarditis. Larger studies are needed to define specific imaging findings of this rare condition. Endomyocardial biopsy (EMB) confirms the diagnosis of myocarditis and identifies the underlying etiology and the type of inflammation (e.g., giant cell, eosinophilic myocarditis, sarcoidosis), which imply different treatments and prognosis. The European Society of Cardiology Working Group on Myocardial and Pericardial Diseases gave EBM the highest levels of recommendations in the life-threatening clinical presentation, which was not the current case [8]. The procedure of EBM depends upon local availability and expertise. Thus, it was reasonable to first perform CMR in clinically stable patients, prior to EMB.

An etiological infectious workup was performed, according to the European cardiac recommendation. The viral serology is of limited utility in the diagnosis of myocarditis because it does not imply myocardial infection but rather the interaction of the peripheral immune system with infectious agent. In the North American series of myocarditis, molecular techniques, mainly (reverse transcriptase) RT-PCR amplification, showed genomes of enterovirus, adenovirus, influenza viruses, human herpes virus-6 (HHV-6), Epstein-Barr-virus, cytomegalovirus, hepatitis C virus and parvovirus B19 in the myocardium of patients with myocarditis [10]. This approach is not consensual. In fact, the diagnostic value of EBM was based of the Dallas histopathologic criteria and did not include viral genome analysis.

The recent scientific statement on EMB gave the highest levels of recommendation in life-threatening clinical presentations. However, the diagnostic, prognostic and therapeutic value of EMB was based on the Dallas histopathologic criteria and did not include immunohistochemistry and viral genome analysis [11].

Further studies are needed to establish the real link between the viral agents and their cardio tropism. The place of RT-PCR amplification in the myocardium is to be determined.

*Campylobacter* spp. have an uncommon but increasingly recognized link with myocarditis, as supported by a growing number of case reports and reviews of literature [4,5,6]. *C. fetus* is less common as a human pathogen [1] but known to have higher predilection for heart and vascular endothelium, specifically causing endocarditis [12]. The putative mechanisms of myocarditis are not well elucidated. They include several mechanisms, such as direct infection of myocardium, an immune-mediated response or the effect of bacterial toxins. Given the short gap between the onset of enteric and cardiac symptoms, contrary to those seen with *C. jejuni* and development of other immune-mediated diseases (e.g., Guillain-Barré syndrome, reactive arthritis), auto immune mechanism seems not to be the dominant mechanism in the myocardial lesions due to *C. jejuni* [13]. The patient received conservative treatment associated with antibiotics. Previous reports highlighted the place of non-steroidal anti-inflammatory drugs and colchicine. In our case, these therapeutics were not used probably because we did not observe associated pericarditis [5]. In addition, the association of colchicine and macrolides in a situation of acute cardiac damage could induce arrhythmia. The global outcome was favorable in this case, in agreement with low rates of morbidity, mortality, evolution to heart failure and worsening ventricular function [14].

## Figures and Tables

**Figure 1 microorganisms-10-00208-f001:**
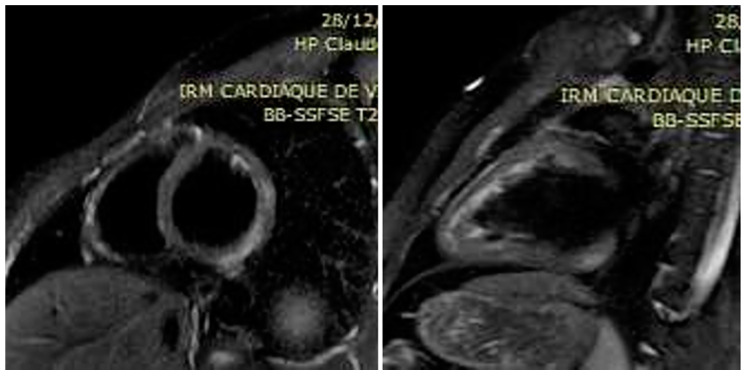
Black-blood edema-sensitive T2 STIR images showing subepicardial posterolateral hypersignal (arrows) indicative of focal edema in the short axis and the 2-chamber view of the left ventricle.

**Figure 2 microorganisms-10-00208-f002:**
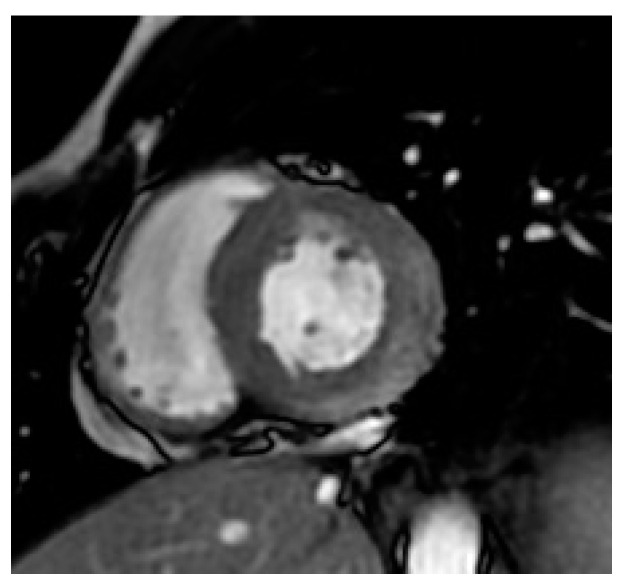
End-systolic image extracted from SSFP cine CMR in the basal LV short-axis view, showing early hypersignal in the subepicardium of the posterolateral wall shortly after (1 min.) injection of 0.1 mM of Gadolinium chelates, indicating focal hyperemia (arrows).

**Figure 3 microorganisms-10-00208-f003:**
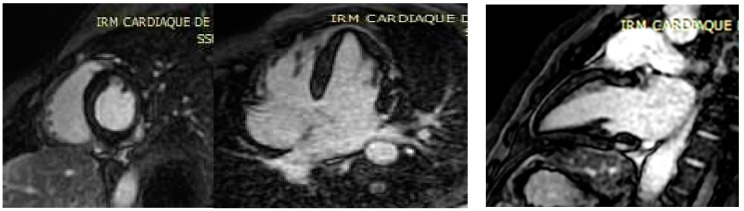
Inversion-recovery gradient-echo-based late Gadolinium enhancement images, acquired 10 min. after Gadolinium injection, showing subepicardial nodular lesions of myocardial damage (arrows).

## Data Availability

All data and material collected during this study are available from the corresponding author upon reasonable request.

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
