# Peer review of "Cardiovascular Magnetic Resonance Imaging Pattern in Campylobacter jejuni-related Myocarditis"

_microorganisms, 2022, doi:10.3390/microorganisms10020208_

Round 1

Reviewer 1 Report

I read this case report with great interest. A relatively comprehensive description on a clinical suspected myocarditis case caused by Campylobacter Jejuni (C. jejuni).

The major point on this case is lacking of biopsy confirmation for the diagnosis. The clinical course of the case is match also with transient myocarditis caused virus infection.  Noting the relative mild clinical manifestation, it is reasonable to avoiding an invasive operation, but from a rigorous scientific point of view, it should be excluded possible infection of cardiophilia virus in heart specimen. The authors checked only serological RNA and DNA viruses, but this cannot represent the reality in heart, and vice inverse.

The minor point is that the authors did not repeat cardiac MRI after clinical recovery, which might help to identify if any insistent fibrosis and might associate with prognosis regarding a long term of time.

Author Response

Souheil Zayet MD

Infectious disease department

Nord Franche Comté hospital 9000 Belfort, France

souhail.zayet@gmail.com

Subject : reply to reviewer’s comments

Dear Editor,

Dear Sir or Madam,

We are thankful for your efforts in reviewing our manuscript and glad that you find our work of great interest.

Thus, we read carefully your remarks and had modified our text according to your comments.

Hereby, our reply.

Reviewer 1 :

It is certain that EMB is the gold standard diagnostic tool to establish the diagnosis of myocarditis. In our case, EBM was not performed because local team is not experienced with such procedure. Moreover, recommendations stipulated that EBM should be proposed in the life-threatning clinical presentation, which was not the current case.

According to reviewers comments, we detailed this comments in the discussion part.

Moreover, a large viral serology workup was performed according to the European society of cardiology recommendations. According, to reviewer 1 comments, we highlight that the serology’s are of poor aid and do not prove their implication in myocardial injury. Previous published data showed that viral RT PCR in the myocardium tissue were found but this approach is not consensual. In fact, the diagnostic value of EBM was based of the Dallas histopathologic criteria and did not include viral genomes analysis.

We are looking forward for your reply.

Yours faithfully,

Souheil Zayet MD

Reviewer 2 Report

This is a case report of myocarditis caused by C. jejuni, and is unique in that it mentions MRI findings.

This is a very interesting case. However, it needs some modifications which are described below.

Abstract

No special modifications are required.

Introduction

No special modifications are required.

Case Presentation

Please present his respiratory rate.

Please present a 12-lead ECG.

Please provide a numerical value for the WBC and its fractions.

Indicate whether human herpes virus-6, Epstein-Barr virus, varicella-zoster virus, and herpes simplex virus are in a previously infected or uninfected pattern. Antibody titer should be clearly indicated.

Please mention maximum CPK and troponin levels and changes over time.

Present the echocardiographic findings.

Discussion

Describe the putative mechanism by which C. jejuni causes myocarditis.

Author Response

Souheil Zayet MD

Infectious disease department

Nord Franche Comté hospital 9000 Belfort, France

souhail.zayet@gmail.com

Subject : reply to reviewer’s comments

Dear Editor,

Dear Sir or Madam,

We are thankful for your efforts in reviewing our manuscript and glad that you find our work of great interest.

Thus, we read carefully your remarks and had modified our text according to your comments.

Hereby, our reply.

Reviewer 2 :

Modifications were performed in the clinical presentation according to reviewer’s comments.

The putative mechanisms of myocarditis are not well elucidated. We detailed the reported mechanism in the literature in the discussion part.

We are looking forward for your reply.

Yours faithfully,

Souheil Zayet MD

Round 2

Reviewer 2 Report

It can be accepted.